# Age-Specific Clinical and Laboratory Features and Renal Involvement in Children with MIS-C: A Single Tertiary Centre Experience from Vojvodina

**DOI:** 10.3390/medicina61071142

**Published:** 2025-06-25

**Authors:** Borko Milanović, Vesna Stojanović, Gordana Vijatov-Ðurić, Marijana Savin, Andrea Ðuretić, Jelena Kesić, Nenad Barišić, Ognjen Ležakov, Ivana Vorgučin, Gordana Vilotijević-Dautović, Katarina Koprivšek

**Affiliations:** 1Medical Faculty of Novi Sad, University of Novi Sad, Hajduk Veljkova 3, 21000 Novi Sad, Serbia; borko.milanovic@mf.uns.ac.rs (B.M.); vesna.stojanovic@mf.uns.ac.rs (V.S.); gordana.vijatov-djuric@mf.uns.ac.rs (G.V.-Ð.); marijana.savin@mf.uns.ac.rs (M.S.); jelena.kesic@mf.uns.ac.rs (J.K.); nenad.barisic@mf.uns.ac.rs (N.B.); ivana.vorgucin@mf.uns.ac.rs (I.V.); gordana.vilotijevic-dautovic@mf.uns.ac.rs (G.V.-D.); katarina.koprivsek@mf.uns.ac.rs (K.K.); 2Institute for Child and Youth Healthcare of Vojvodina, Hajduk Veljkova 10, 21000 Novi Sad, Serbia; andrea.djuretic@izzzdiovns.rs

**Keywords:** COVID-19, MIS-C, multisystem inflammatory syndrome, SARS-CoV-2, child, kidney diseases, acute kidney injury, anti-inflammatory agents, acetylsalicylic acid, methylprednisolone

## Abstract

*Backgrounds and Objectives*: Multisystem Inflammatory Syndrome in Children (MIS-C) is a rare but potentially severe complication of SARS-CoV-2 infection, with increasingly reported renal manifestations. *Materials and Methods*: The aim of this retrospective study was to compare clinical and laboratory characteristics across age categories, with special emphasis on renal function. We analysed data from 64 patients with MIS-C treated between July 2020 and December 2023. *Results*: In children under 3 years of age, there was a higher prevalence of leucocytosis, elevated platelet counts, and anaemia, along with a lower frequency of complications. The 3–6-year age group was characterized by the presence of rash, hypoalbuminemia, and elevated transaminases. The 7–12-year age group showed the highest rate of organ dysfunction. In adolescents (13–18 years), neurological symptoms, the highest BMI values, the greatest prevalence of comorbidities, leukopenia, lymphopenia, and elevated GGT levels were observed. The incidence of acute kidney injury (AKI) was 6.3% (*n* = 4/64). Following treatment, the majority of patients achieved full recovery (*n* = 61/64; 95.2%). *Conclusions*: There are pronounced age-related differences in the clinical presentation of MIS-C, with distinct immune and clinical patterns suggesting developmental influences on disease expression and outcomes. Older children showed a higher prevalence of comorbidities and organ dysfunction compared to younger patients. Notably, this study found a markedly lower incidence of acute kidney injury (6.3%) compared to previously reported rates (20–30%), indicating potential regional or age-related protective factors. These findings highlight the importance of age-specific evaluation in MIS-C and underscore the need for further multicentre research to refine therapeutic protocols.

## 1. Introduction

Coronavirus disease 2019 (COVID-19) in children usually presents with mild symptoms or remains asymptomatic. However, 2–6 weeks after infection, a small proportion of children may develop a severe, life-threatening condition known as Multisystem Inflammatory Syndrome in Children (MIS-C) [1]. The incidence of MIS-C is approximately 1 case per 3000 SARS-CoV-2 infections or 5.1 per 1,000,000 person-months in individuals under 21 years of age [2]. MIS-C results from hyperinflammation, endotoxemia, and redox imbalance in genetically predisposed children [3]. Its pathophysiology is linked to intense activation of the innate immune system, uncontrolled cytokine production, and dysfunction of T and B lymphocytes. MIS-C is considered a post-infectious immune-mediated reaction following SARS-CoV-2 infection, possibly involving super-antigen effects and autoantibody formation targeting specific organs [4]. The clinical presentation depends on the degree of inflammation and includes multiorgan dysfunction involving the lungs, gastrointestinal tract, skin, central nervous system, and kidneys. In the most severe cases, cardiovascular involvement, shock, and death may occur [3]. The variability of clinical manifestations and laboratory findings in MIS-C poses a significant differential diagnostic challenge against conditions such as toxic shock syndrome, Kawasaki disease, secondary hemophagocytic lymphohistiocytosis, and macrophage activation syndrome, among others. The history of SARS-CoV-2 infection remains the key differential diagnostic criterion [5]. Although renal involvement is not a dominant feature of MIS-C, an increasing number of studies have reported that acute kidney injury (AKI) is a common and significant complication, occurring in about 20–30% of cases, and is considered a predictive factor for mortality [6,7,8]. In the study by Tripathi et al., one-fifth of children with MIS-C developed AKI and had 4.68 times higher odds of death compared to those without AKI. The mortality rate among MIS-C patients with AKI was comparable to that of paediatric intensive care unit (PICU) patients with AKI due to other causes. Interestingly, their need for kidney replacement therapy (KRT) appeared to be slightly lower, at around 15% [6].

The pathogenesis of renal injury in MIS-C involves hypoperfusion, hyperimmune responses, and also cardiac and endothelial dysfunction. In addition to AKI, haematuria, proteinuria, pyuria, and tubular dysfunction are frequently observed in MIS-C [9,10]. The therapeutic protocol for MIS-C, as defined by the American College of Rheumatology (ACR), includes the initial use of high-dose intravenous immunoglobulin (IVIG) and methylprednisolone, with pulse doses of methylprednisolone and biological agents (anakinra and infliximab) reserved for refractory cases [11].

While the incidence and clinical spectrum of MIS-C have been increasingly described, significant gaps remain in understanding age-related differences in disease expression and renal involvement. Although previous studies have reported a high incidence of acute kidney injury, data from Central and Eastern Europe are scarce. It remains unclear whether regional, genetic, or developmental factors may influence disease severity and organ-specific complications. The aim of this study was to compare the clinical and laboratory characteristics of patients with MIS-C across different age groups, with a special focus on renal function.

## 2. Materials and Methods

This retrospective study included all patients aged 0 to 18 years diagnosed with MIS-C and treated at the Institute for Child and Youth Health Care of Vojvodina from July 2020 to December 2023, with a one-year follow-up post-discharge. The Institute for Child and Youth Health Care of Vojvodina is the only paediatric facility with a Paediatrics Intensive Care Unit (PICU) in the Autonomous Province of Vojvodina and the only institution that managed paediatric MIS-C cases in the region. MIS-C diagnosis was established according to the CDC and WHO criteria [12,13]. Parents or legal guardians of all participants provided informed consent for hospital admission and procedures performed during hospitalization. Data were retrospectively collected from medical records. Evidence of SARS-CoV-2 infection was confirmed by a positive SARS-CoV-2 antigen test or a positive reverse transcription polymerase chain reaction (RT-PCR) test from oro/nasopharyngeal swabs upon admission, positive serology (IgM and/or IgG against SARS-CoV-2), or a documented recent infection in the patient or a family member.

Collected data included demographic (age, sex) and anthropometric parameters (body weight, height, body mass index [BMI]), comorbidities, clinical features (presence and duration of fever, Kawasaki-like symptoms, respiratory, gastrointestinal, neurological, and other manifestations), and laboratory findings, considering age-specific reference ranges. Overweight was defined as a body mass index (BMI) ≥ 85th and <95th percentile, while obesity was defined as a BMI ≥ 95th percentile for the corresponding age and sex. Laboratory tests included blood counts, inflammatory markers, cardiac biomarkers, transaminases, coagulation status, renal function parameters, and urinalysis. Procalcitonin (PCT), interleukin-6 (IL-6), pro-BNP, troponin levels, and 24 h proteinuria were not available for all patients due to technical limitations. Pathological 24 h proteinuria was defined as >150 mg/24h. Imaging studies included lung ultrasound, echocardiography, and abdominal ultrasound. Data regarding the administered therapies and PICU admissions were also collected. Indications for PICU admission included the need for mechanical ventilation and hemodynamic instability.

Hospitalization complications and one-year follow-up outcomes were analysed. Acute kidney injury was defined according to KDIGO guidelines, as an increase in serum creatinine of ≥26.5 µmol/L within 48 h or >50% compared to baseline values within 7 days [14]. Patients were treated according to the recommended protocols by the American College of Rheumatology (ACR) [11]. The study was approved by the Institute’s Ethics Committee and conducted in accordance with the principles of the Declaration of Helsinki. Methodology and data reporting adhered to the Strengthening the Reporting of Observational Studies in Epidemiology (STROBE) guidelines [15].

### Statistical Analysis

Data entry and analysis were performed using IBM SPSS Statistics 18.0. Numerical data were presented as frequencies, percentages, means with standard deviations, and minimum and maximum values. Student’s t-test was used to compare two independent samples, and one-way analysis of variance (ANOVA) was applied for comparisons among multiple groups. The chi-square (χ^2^) test was used to assess differences in categorical variables. The Kruskal–Wallis test was used for comparing median values across multiple groups when data were non-parametric. Correlations between two parameters were assessed using Pearson’s correlation coefficient. In order to examine the connection between two or more features, and generate adequate statistical models to assess the impact of observed predictors, univariate binary regression analysis was used. Statistical significance was defined as *p* < 0.05, and high statistical significance as *p* < 0.01.

## 3. Results

### 3.1. Participants

A total of 64 participants were included in the study. For 43.8% of children, data on the timing of SARS-CoV-2 infection were unavailable. In one-fifth of cases (20.3%), infection was confirmed in other family members, while in slightly more than one-third of the children (35.9%), SARS-CoV-2 infection was directly confirmed. The participants were aged between 4 months and 17 years and 8 months, with a mean age of 8 years and 2 months. The participants were divided into four age groups: 0–2, 3–6, 7–12, and 13–18 years. The largest number of participants were aged 7–12 years (*n* = 25/64; 39.1%). The majority of participants were male (*n* = 41/64; 64.1%). There was no statistically significant difference in gender distribution across the different age categories. The mean BMI percentiles were higher among children aged 13–18 years (BMI percentile = 71.82; SD = 28.11) compared to the other age groups, with the difference being highly statistically significant (*p* = 0.008). A total of 57.8% (*n* = 37/64) of patients had no comorbidities. The most common comorbidities were obesity and overweight (each 9.4%, *n* = 6/64), Hashimoto’s thyroiditis (7.8%, *n* = 5/64), malnutrition (6.3%, *n* = 4/64), and asthma (3.1%). Comorbidities were more frequent in the 13–18-years-old group (*n* = 10/64; 15.6%) compared to other age groups, with the difference being statistically significant (*p* = 0.046).

### 3.2. Clinical Characteristics

Neurological symptoms (headache, meningism, seizures) were present in approximately one-fifth of the patients (20.4%), with headache being the predominant complaint. Statistically significant differences were observed in the distribution of neurological symptoms across the age groups, most prevalent in the 13–18-years-old group (*p* = 0.017). A rash was present in less than half of the participants (*n* = 29/64, 45.3%), and it was significantly more common in children aged 3–6 years compared to other age groups (*p* = 0.029). Hypotension was recorded in four children (6.2%). Five children (7.8%) required treatment in the intensive care unit. Clinical characteristics of MIS-C patients according to age categories are presented in Table 1.

### 3.3. Laboratory Data

#### 3.3.1. Laboratory Analyses According to Age Categories

More than half of participants (54.6%) had leucocytosis, and every third participant (34.3%) had lymphopenia. Lymphopenia was statistically significantly more common in the 13–18 years age group. The highest incidence of anaemia was observed in the 0–2 years age group. Mean platelet counts were statistically significantly higher in the 0–2 years age group compared to other groups (*p* = 0.019). In the 3–6 years age group, significantly lower albumin values were observed compared to other groups (*p* = 0.003), and hypoalbuminemia was more frequent in children younger than six years (*p* = 0.001). Increased GGT levels were statistically significantly more common in the 13–18 years age group (*p* = 0.008). Elevated AST and ALT levels were significantly more frequent in the 3–6 and 7–12 years groups. There were no significant differences between age groups regarding LDH, amylase, lipase, or uric acid levels. Elevated troponin levels were found in 63.3% of participants, and elevated pro-BNP levels in 61.5%. There were no statistically significant differences in the distribution of cardiac markers or in coagulation parameters (PT, aPTT, fibrinogen—elevated in 90.6% of participants) among different age groups. Elevated D-dimer values were found in 96.8% (*n* = 61/64) of participants, with mean values of 1637.62 ng/mL. Elevated CRP levels were found in 98.4% (*n* = 63/64), with mean values of 177.39 mg/L. Elevated PCT levels were observed in 91.9% (*n* = 34/37) with a mean value of 12.03 ng/mL, and elevated IL-6 levels were found in all participants, with a mean value of 323.42 pg/mL. There were no statistically significant differences in CRP, PCT, IL-6, or D-dimer values across the age groups. The most common organ dysfunctions were pancreatitis (21,8%, *n* = 14/64), pericarditis (*n* = 11/64; 17.2%), heart failure (7.8%, *n* = 5/64), and acute kidney injury (AKI) (6.3%, *n* = 4/64). Organ dysfunctions were statistically significantly more frequent in the 7–12 years age group compared to the others (*p* = 0.026). The results of laboratory analyses and organ dysfunctions are presented in Table 2.

#### 3.3.2. Renal Function Parameters and AKI

The mean value of 24 h proteinuria was 270 mg/24 h, with a maximum value of 740 mg/24 h. A total of four patients (6.3%) developed AKI. All four patients were male. Three patients with AKI were aged 7–12 years (one aged 7 years and two aged 10 years), and one patient was 14 years old. Renal function parameters and AKI according to age categories are shown in Table 3. Using univariate binary logistic regression analysis, we examined the influence of predictors on the occurrence of AKI. In univariate analysis, the effect of each individual predictor was assessed separately. The only statistically significant predictor of AKI was elevated uric acid with a *p*-value of 0.044 and an odds ratio (OR) of 9.000, meaning patients with elevated uric acid levels have a nine-fold increased likelihood of developing AKI compared to those with normal uric acid levels.

### 3.4. Radiography and Imaging Data

Pathological abdominal ultrasound findings were most common in the 3–6 years age group compared to others, and the differences were statistically significant (*p* = 0.014). Abnormal abdominal ultrasound findings included splenomegaly (*n* = 19/64; 29.7%), ascites (*n* = 9/64; 14%), thickening of intestinal loops (*n* = 12/64; 18.75%), and enlarged mesenteric lymph nodes (*n* = 4/64; 6.25%). There was no statistically significant difference in lung ultrasound findings among age groups; findings included pleural effusion (*n* = 20/64; 31.25%), pulmonary parenchymal condensation (*n* = 4/64; 6.25%), and interstitial oedema (*n* = 2/64; 3.1%). Echocardiography showed pericarditis (*n* = 11/64; 17.2%), heart failure (*n* = 5/64; 7.8%), tricuspid valve insufficiency (*n* = 3/64; 4.6%), and mitral valve insufficiency (*n* = 3/64; 4.6%). There was no statistically significant difference in echocardiography findings across the age groups.

### 3.5. Therapy

Data on applied therapy are presented in Table 4. There were no statistically significant differences in the mean dose of methylprednisolone or acetylsalicylic acid (ASA) across age groups.

There was no statistically significant association between the occurrence of AKI and the dose of acetylsalicylic acid (t = 1.898, df = 57, *p* = 0.063) or methylprednisolone (t = 0.123, df = 59, *p* = 0.902). Since all four AKI cases received ASA dosage of 3 mg, the resulting standard deviation (SD) was 0.00. The results are presented in Table 5.

### 3.6. Long-Term Outcomes

Most patients achieved full recovery after treatment (*n* = 61/64; 95.2%). One patient had mild persistent proteinuria (0.17 g/24 h, 24 h microalbuminuria 30.8 mg/24 h), and another had persistently elevated amylase and lipase levels. There was one fatal outcome (1.6%).

## 4. Discussion

Multisystem Inflammatory Syndrome in Children (MIS-C) is a rare but potentially life-threatening complication of SARS-CoV-2 infection. Previous studies suggest a higher prevalence among younger school-aged children and males, but the role of age as a risk factor for developing more severe forms of MIS-C remains insufficiently explored [16,17,18]. In our study, which included 64 children treated for MIS-C at the Institute for Child and Youth Healthcare of Vojvodina, we analysed clinical, laboratory, radiological, and therapeutic characteristics across different age groups, with a special focus on renal function parameters. The average age of children with MIS-C was 8 years and 2 months, and most participants were male, consistent with previous research [16,17,18,19]. The most common symptoms were general and gastrointestinal, with respiratory symptoms present in half of the participants. These findings align with the clinical symptom distribution reported by Hoste et al., except for the incidence of severe clinical presentation with shock (56.3%) and the need for intensive care and therapy (73.3%), which was higher in their study than in ours [17]. MIS-C is characterized by certain hematologic abnormalities such as lymphopenia, neutrophilia, thrombocytopenia, and anaemia [20]. In the age group under three years, a higher incidence of leucocytosis and anaemia, as well as higher platelet counts, were observed, possibly reflecting both an acute inflammatory response and developmental characteristics specific to younger children. This group also showed the lowest incidence of lymphopenia, likely due to the physiological lymphocytosis typical for this age as part of normal immune system maturation [21]. Higher average platelet counts in this group may indicate reactive thrombocytosis. At this developmental stage, the innate immune response predominates, along with increased IL-6 production, which stimulates platelet synthesis via thrombopoietin [22,23]. The higher prevalence of anaemia in children under three may result from increased iron requirements during rapid growth, but also from inflammation-induced anaemia, characterized by functional iron deficiency due to increased hepcidin production, ferroprotein inhibition, reduced iron absorption, and oxidative stress [24]. This age group tends to develop complications less frequently, and their systemic inflammatory response may manifest through specific hematologic patterns reflecting both the physiological development of the immune and hematopoietic systems and a reactive response to inflammation within MIS-C. In the 3–6-years-old group, significantly lower albumin levels were noted. Hypoalbuminemia is a common laboratory finding in MIS-C. At this age, children are undergoing an immunological transition, with decreasing innate immunity dominance and not yet fully matured adaptive immunity, potentially contributing to inadequate inflammatory regulation [22]. Increased capillary permeability (“vascular leak”) under the influence of pro-inflammatory cytokines, particularly IL-6, leads to albumin loss from the intravascular space [25]. Additionally, albumin is a negative acute-phase reactant, meaning its synthesis decreases during inflammation, as described in other MIS-C studies [20]. Elevated ALT and AST levels were also statistically significantly more common in the 3–6-years-old group (AST also in 7–12-year-olds), potentially indicating liver synthetic dysfunction contributing to hypoalbuminemia. The 3–6-years-old group also showed the highest prevalence of abnormal abdominal ultrasound findings (splenomegaly, ascites, intestinal wall thickening, enlarged mesenteric lymph nodes) and the highest rate of rash, which in MIS-C can appear as maculopapular, urticarial, or vesicular eruptions [26]. The 7–12-years-old group emerged as the most vulnerable to developing severe clinical presentations, with the highest rates of organ dysfunction (e.g., pancreatitis, pericarditis, AKI, heart failure) and also the largest proportion of patients overall. Our findings are consistent with the multicentre study by Feldstein et al., where children aged 6–12 accounted for the majority of MIS-C cases with prominent multiorgan involvement and cardiac dysfunction [27]. In our study, patients who developed AKI were aged 7, 10 (2 patients), and 14 years. A study by Lipton et al. investigating AKI in MIS-C patients aged 0–20 years found that those who developed AKI had significantly higher median ages compared to those without AKI. One possible explanation is that younger children were brought in for medical evaluation earlier, allowing for timely intervention and reduced risk of renal injury progression [28]. This raises the question—why is this age group most at risk? A potential explanation involves a combination of immunologic, hormonal, and socio-epidemiological factors. In this age group, there is an imbalance in immune regulation (T-cell regulatory functions, cytokine responses, lack of prior immunologic memory), leading to a more intense immune response and increased risk of severe MIS-C [29,30]. Additionally, school attendance increases exposure to viruses, including SARS-CoV-2, potentially provoking a more robust immune activation. Studies have shown that children with MIS-C have distinct T-cell responses compared to those who recover from COVID-19 without complications, including increased frequencies of virus-specific proinflammatory memory T cells [31]. Hormonal changes during prepubertal or pubertal phases (also affecting this age group) could further influence immune responses, given that oestrogens and androgens are known to modulate immunity and inflammation [32]. In the 13–18-years-old group, the highest prevalence of neurological symptoms, highest BMI percentiles, greatest comorbidity presence, leukopenia, lymphopenia, and elevated GGT levels were observed. Neurological symptoms predominating in this group suggest a possible age-specific neuroinflammatory pattern. No similar data were found in the available literature. Our results align with a large population-based study by Rhedin et al., where obesity and asthma were most common in the 12–18-year MIS-C population [18]. Adipose tissue acts as an endocrine organ producing proinflammatory cytokines, contributing to inflammation and endothelial dysfunction, negatively impacting the entire organism [10,33]. Rhedin et al. also highlighted obesity as a significant risk factor for MIS-C development [18]. Elevated GGT levels, as a marker of inflammation and bile duct injury, were statistically more common in 13–18-year-olds. Elevated GGT often accompanies obesity as part of metabolic syndrome, aligning with our findings of the highest BMI percentiles in this age group [34]. Lymphopenia, a known finding in MIS-C often associated with severe disease, and leukopenia, possibly due to intense inflammation or cytokine-driven hematopoietic suppression, were more frequent in this group. Chronic adipose tissue inflammation may further exhaust the immune system, reflecting higher rates of leukopenia and lymphopenia [35,36]. According to a 2022 meta-analysis by Tripathi et al., overall mortality among MIS-C patients was 4%. Mortality increased fivefold in those who developed AKI [6]. In our study, there was one fatality (1.6%). Additionally, when comparing AKI prevalence with the available literature, we observed a significantly lower rate of renal impairment—only 6.3% (four participants), about 4–5 times lower than previously reported studies [7,8]. Despite a pronounced inflammatory response among most patients, renal function remained preserved in the majority. The aetiology of MIS-C is multifactorial, with potential contributors including hypoperfusion, hyperimmune inflammatory responses, and hypercoagulability leading to microthrombi [37]. Adequate suppression of inflammation contributes to hemodynamic stability and renal perfusion. In MIS-C-related renal injury pathophysiology, initial renal ischemia triggers an inflammatory cascade through chemokine and cytokine production (IL-1, IL-6, IL-8, MCP-1, RANTES, TNF-α), leading to continued inflammation even after the ischemic event resolves, resulting in tubular cell death and reduced glomerular filtration [10,38]. SARS-CoV-2 antigens have been immunohistochemically detected in renal tubular epithelial cells and podocytes. MIS-C renal biopsy findings indicate combined glomerular and tubular injury, including thrombotic microangiopathy, focal acute tubular necrosis, and IgM deposition, suggesting an immunologic injury mechanism [39]. Diorio et al. provided important insights into the pathophysiology of renal injury in MIS-C, suggesting that kidney damage may result from a combination of direct viral infection, complement activation, and vascular injury. Their study demonstrated that elevations in soluble C5b-9 (sC5b9), a marker of terminal complement activation, were independently associated with renal dysfunction. Notably, sC5b9 levels correlated with markers of renal impairment, but were not linked to typical laboratory features of thrombotic microangiopathy (TMA), such as elevated LDH or decreased platelet and haemoglobin levels. Furthermore, sC5b9 elevations occurred independently of other inflammatory markers, highlighting complement activation as a distinct and possibly central mechanism of injury. The presence of elevated sC5b9 in all paediatric SARS-CoV-2-related illness groups, compared to healthy controls, further suggests that complement activation and subclinical TMA are prevalent in children with COVID-19 and MIS-C [40]. Treatment with IVIG and glucocorticoids, through their immunomodulatory effects, reduces proinflammatory cytokine synthesis (IL-1β, IL-6, TNF-α), thereby lowering systemic inflammation, stabilizing vascular permeability, and maintaining renal perfusion, potentially decreasing AKI risk [3,22].

In a multivariate analysis conducted by Grewal et al., who were investigating AKI in acute SARS CoV2 infection and MIS-C, the most significant predictors for AKI were requirement of inotropes and African-American race [41]. In our case, the only predictor was elevated uric acid levels, which is an expected laboratory finding in AKI development. It is possible there are other predictors for AKI, which was not shown in our paper, considering only four patients developed AKI. The combined effect of inflammation reduction and perfusion preservation may contribute to the preservation of renal function in MIS-C. When comparing ASA dosing and AKI incidence, a borderline statistical significance (*p* = 0.063) was observed; patients without AKI received an average ASA dose of 3.89 mg/kg (SD 0.93) compared to 3 mg/kg (SD 0.00) with AKI. These findings suggest that higher ASA doses may have a potential nephroprotective effect, but this should be interpreted with caution, considering the small number of AKI cases (*n* = 4). No studies to date have investigated the impact of ASA on renal function in MIS-C. ASA’s potential nephroprotection could be linked to MIS-C’s hypercoagulable state, with coagulopathy and microthrombi formation underlying renal injury in various conditions [38,42]. Given the multifactorial nature of MIS-C pathophysiology, including systemic inflammation, hemodynamic instability, hypotension, and capillary leak, we cannot establish a causal relationship between ASA dosing and renal outcomes. However, our findings highlight the need for further research to explore a potential nephroprotective role of antiplatelet therapy in this context. Pathological 24 h proteinuria was detected in 14 out of 23 patients tested (60.9%), with an average 24 h proteinuria of 270 mg. In a study by Meneghel et al. analysing tubular dysfunction in MIS-C, a lower proteinuria incidence of 34% (9/26 patients with 24 h samples) was noted. Additionally, in their study, two cases of nephrotic proteinuria (>50 mg/kg/day) were recorded, which was not observed in our investigation. However, in their study, acute kidney injury (AKI) was noted in 23% of patients, which is 3.5 times higher than our results [37]. These differences could be explained by potentially different clinical criteria for selecting patients for 24 h analysis, differences in the severity of the clinical presentation, and the timing of urine sampling in relation to the administration of therapy. Elevated body temperature, which was present in all patients, is a known cause of transient proteinuria, so it is necessary to differentiate proteinuria caused by fever from that directly associated with MIS-C. In our study, the majority of children showed no residual kidney damage upon discharge and during one-year follow-up, except for mild persistent proteinuria in one patient. This result is in line with the study by El-Halaby et al., who reported the complete recovery of kidney function in all MIS-C patients by the time of discharge and during a six-month follow-up [9]. However, our findings differ from the study by Kari et al., where residual kidney damage, without urine analysis, was verified in 9% of patients after confirmed COVID-19 infection, including those with MIS-C. They identified factors associated with residual damage, such as reduced tissue perfusion, sepsis, worsening clinical condition, or the presence of comorbidities [43]. Potential reasons for these differences could be early disease recognition, timely initiation of intensive therapy, a lower frequency of comorbidities in our sample, or differences in the severity of clinical presentation.

Despite the proposed mechanisms of injury, our cohort exhibited a notably low incidence of AKI, relatively mild proteinuria, and no residual kidney damage in the majority of children during follow-up. Several protective factors may explain these findings. Early administration of anti-inflammatory therapy (IVIG and corticosteroids) likely attenuated the cytokine-mediated inflammatory cascade, preserving endothelial integrity and renal perfusion [3,22]. Hemodynamic stability was maintained in most patients, thereby preventing hypoperfusion-related ischemic injury [10,37,38]. Additionally, the lower prevalence of comorbidities such as obesity—commonly associated with endothelial dysfunction and proinflammatory states—may have contributed to better renal outcomes [10,33]. Together, these factors may have acted synergistically to protect renal function in our patients despite the presence of systemic inflammation.

### Limitations of the Study

Our study has several limitations. First, it presents the results of a single tertiary care centre. Although the uniformity in data collection is a strength, the relatively small sample size represents a limitation. In addition, the retrospective nature of data collection imposes its own constraints. The small number of patients with AKI in our cohort highlights the need for further research with larger samples to more accurately assess the incidence and characteristics of kidney injury in MIS-C. Furthermore, as the clinical manifestations and severity of MIS-C may vary significantly across different racial groups, it is important to note that this study included only patients of similar racial backgrounds. Finally, COVID-19 vaccination status was not considered in the analysis.

## 5. Conclusions

The results of this study indicate that there are significant age-related differences in the clinical presentation, laboratory findings, and risk of complications in children with MIS-C. Older children have a higher frequency of comorbidities and organ dysfunction compared to younger children with MIS-C. Kidney function remained preserved in most children, with a significantly lower frequency of acute kidney injury compared to earlier studies, indicating potential regional or age-related protective factors. Additional multicentre research is needed to redefine therapeutic protocols for MIS-C.

## Figures and Tables

**Table 1 medicina-61-01142-t001:** Clinical characteristics of MIS-C patients according to age categories.

Characteristic		Total *n* = 64 (100%)	0–2 Years *n* = 15 (23.4%)	3–6 Years *n* = 11 (17.2%)	7–12 Years *n* = 25 (39.1%)	13–18 Years *n* = 13 (20.3%)	*p*	
Fever Mean/SD *		39.45C (SD = 0.67)	39.43C (SD = 0.47)	39.45C (SD = 0.84)	39.60C (SD = 0.58)	39.17C (SD = 0.84)	0.323	
Respiratory symptoms **	Cough (%)	13 (20.3)	3 (20.0)	2 (18.2)	4 (16.0)	4 (30.8)	0.143	
	Pharyngitis, (%)	7 (10.9)	0 (0.0)	2 (18.2)	4 (16.0)	1 (7.7)		
	Rhinorrhoea, (%)	5 (7.8)	2 (13.3)	1 (9.1)	0 (0.0)	2 (15.4)		
	Other (%)	8 (12.5)	3 (20.0)	0 (0.0)	2 (8.0)	3 (23.1)		
GI symptoms **	Vomiting, (%)	33 (51.6)	2 (13.3)	8 (72.7)	17 (68.0)	6 (46.2)	0.407	
	Diarrhoea, (%)	12 (18.8)	9 (60.0)	0 (0.0)	2 (8.0)	1 (7.7)		
	Other (%)	5 (7.8)	0 (0.0)	2 (18.2)	1 (4.0)	2 (15.4)		
Neurological symptoms **	Headache (%)	11(17.2)	0 (0.0)	1 (9.1)	5 (20.0)	5 (38.5)	0.051	0.017
	Meningism (%)	1 (1.6)	0 (0.0)	0 (0.0)	1 (4.0)	0 (0.0)	0.668	
	Seizures (%)	1 (1.6)	0 (0.0)	0 (0.0)	0 (7.7)	1 (1.6)	0.270	
Skin and mucosal lesions	Rash (%) **	29 (45.3)	4(26.7)	9 (81.8)	12 (48.0)	4 (30.8)	0.029	
	Bulbar conjunctivitis, (%) **	22 (34.4)	6 (40.0)	5 (45.5)	9 (36.0)	2 (15.4)	0.412	
	Mucositis, (%) **	23 (35.9)	5 (33.3)	7 (63.6)	9 (36.0)	2 (15.4)	0.112	
Lymphatic system	Cervical lymphadenopathy, (%) **	29 (45.3)	8 (53.3)	7 (63.6)	11 (44.0)	3 (23.1)	0.219	
Extremities **	Feet/Hand oedema, (%) **	8 (12.5)	3 (20.0)	1 (9.1)	4 (16.0)	0 (0.0)	0.395	
	Palmar/Plantar erythema, (%) **	10 (15.6)	3 (20.0)	2 (18.2)	5 (20.0)	0 (0.0)	0.392	

* ANOVA; ** Kruskal–Wallis; BMI: Body Mass Index; GI symptoms: Gastrointestinal symptoms.

**Table 2 medicina-61-01142-t002:** Characteristics of laboratory findings and organ dysfunction according to age categories.

Variable		Total	0–2 Years	3–6 Years	7–12 Years	13–18 Years	*p*
		64 (100%)	*n* = 15 (23.4%)	*n* = 11 (17.2%)	*n* = 25 (39.1%)	*n* = 13 (20.3%)	
Leukocytes	Mean,	12.96	15.46	14.63	11.69	11.08	0.257
(RR 5–10 × 10^9^/L)	SD *	26.22	5.21	7.72	7.93	6.86	
	Leucocytosis, (%) **	33 (51.6)	11 (73.3)	8 (72.7)	9 (36.0)	5 (38.5)	0.046
	Leukopenia, (%) **	2 (3.1)	0 (0.0)	0 (0.0)	0 (0.0)	2 (15.4)	0.047
Neutrophils	Mean,	10.38	10.99	11.9	10.06	9.01	0.698
(RR 2.5–8.5 × 10^9^/L)	SD *	6.28	4.53	6.69	7.21	6.04	
	Neutrophilia (%) **	39 (60.9)	11 (73.3)	9 (81.8)	13 (52.0)	6 (46.2)	0.176
Lymphocytes	Mean, SD*	1.72	3.19	1.65	1.15	1.19	<0.001
(RR 1.3–4.5 × 10^9^/L)		1.28	1.49	0.78	0.69	0.98	
	Lymphopenia (%) **	22 (34.38)	0 (0.0)	1 (9.1)	13 (52.0)	8 (61.5)	<0.001
Haemoglobin	Mean, SD *	108.59	99.33	98.55	109.52	126	<0.001
(RR 100–140 g/L),		17.38	9.46	13.46	16.98	14.94	
	Anaemia (%) **	40 (62.5)	14 (93.3)	10 (90.9)	14 (56.0)	2 (15.4)	<0.001
Platelets	Mean, SD *	296.94	381.73	287.91	287.68	224.54	0.019
(RR 150–400 × 10^9^/L)		136.78	117.23	122.21	124.07	58.47	
	Thrombocytosis (%) **	9 (14.1)	4 (26.7)	1 (9.1)	4 (16.0)	0 (0.0)	0.228
Albumin	Mean, SD *	30.18/5.65	29.26/3.44	27.21/3.07	29.56/4.89	34.91/7.98	0.003
(RR 38–54 g/L)	Hypoalbuminemia, (%) **	56 (87.5)	15 (100.0)	11 (100.0)	23 (92.0)	7 (53.8)	0.001
Gamma–glutamyl transferase	Mean, SD*	0.82/0.81	0.36/0.28	0.72/0.50	1.18/1.07	0.76/0.59	0.016
(RR 0.07–0.37 ukat/L)	Raised GGT, (%) **	41 (64.1)	4 (26.7)	8 (72.7)	19 (76.0)	10 (76.9)	0.008
Aspartate aminotransferase	Mean, SD *	1.09/1.84	0.52/0.19	2.27/4.15	0.99/0.69	0.96/0.86	0.104
(RR 0.08–0.6 ukat/L)	Raised AST, (%) **	32 (50.0)	3 (20.0)	7 (63.6)	16 (64.0)	6 (46.2)	0.043
Alanine aminotransferase	Mean, SD *	0.91/1.41	0.35/0.29	1.72/3.03	0.80/0.59	1.07/0.98	0.094
(RR 0.20–0.98 ukat/L)	Raised ALT, (%) **	25 (39.1)	1 (6.7)	6 (54.5)	12 (48.8)	6 (46.2)	0.034
Lipase ***	Mean, SD *	2.73/ 2.93	1.35/ 0.21	1.81/ 0.86	2.27/ 1.88	4.39/ 4.68	0.355
	Elevated blood lipase **	18/23 (78.3)	2/2 (100.0)	3/4 (75.0)	7/10 (70.0)	6/7 (85.7)	0.763
Amylase ****	Mean, SD *	2.88/ 2.41	1.25/ 0.70	3.16/ 0.68	2.99/ 2.61	3.01/ 3.10	0.820
	Elevated blood amylase **	16/24 (66.7)	0/2 (0.0)	4/4 (100.0)	8/11 (72.7)	4/7 (57.1)	0.102
Organ dysfunction		25 (39.1)	2 (13.3)	3 (27.3)	15 (60.0)	5 (38.5)	0.026

* ANOVA; ** Kruskal–Wallis. *** Based on a sample of 23 participants. **** Based on a sample of 24 participants. RR: Reference ranges. GGT: Gamma–Glutamyl Transferase. AST: Aspartate Aminotransferase. ALT: Alanine Aminotransferase.

**Table 3 medicina-61-01142-t003:** Renal function parameters and incidence of acute kidney injury (AKI) across age groups.

Variable		Total	0–2 Years	3–6 Years	7–12 Years	13–18 Years	*p*
		64 (100%)	*n* = 15 (23.4%)	*n* = 11 (17.2%)	*n* = 25 (39.1%)	*n* = 13 (20.3%)	
Urea	Mean, SD *	4.33/2.40	3.02/1.01	3.58/1.16	5.39/3.09	4.45/1.99	0.012
(RR 2.5–6 mmol/L)	Raised urea, (%) **	7 (10.9)	0 (0.0)	0 (0.0)	6 (24.0)	1 (7.7)	0.055
Creatinine Clearance	Mean, SD *	165.6	168.66	175.78	164.25	155.09	0.842
(ml/min/1.73 m2)		52.56	62.66	42.32	62.09	34.58	
24 h proteinuria g/24 h ***	Mean, SD *	0.27/0.17	0.13/0.17	0.22/0.16	0.29/0.16	0.27/0.2	0.647
	Raised values (%) **	14/23 (60.9)	0/2 (0.0)	1/3 (33.3)	8/10 (80.0)	5/8 (62.5)	0.147
Urine	Sterile pyuria **	4 (6.3)	0 (0.0)	1 (9.1)	2 (8.0)	1 (7.7)	0.727
AKI **		4 (6.3)	0 (0.0)	0 (0.0)	3 (12.0)	1 (7.7)	0.370

* ANOVA; ** Kruskal–Wallis; *** Based on a sample of 23 participants; AKI: Acute Kidney Injury; RR: Reference Ranges.

**Table 4 medicina-61-01142-t004:** Dose, administration frequency, and treatment duration.

Therapy	Frequency (%, *n*)	Mean Dose (Dose Range)	Mean Duration of Therapy
IVIG	100% (64)	2 g/kg	Single administration
Methylprednisolone	95.3% (61)	1.28 mg/kg (1–2 mg/kg)	5.84 weeks
Pulse methylprednisolone	2.56% (4)	23.75 mg/kg (15–30 mg/kg)	3 days
ASA	92.2% (59)	3.83 mg/kg (2–5 mg/kg)	5.95 weeks
Dobutamine	4.7% (3)	–	4 days

IVIG: Intravenous Immunoglobulin; ASA: Acetylsalicylic Acid.

**Table 5 medicina-61-01142-t005:** Average doses of acetylsalicylic acid and methylprednisolone in patients with and without AKI.

Variable	Dose of ASA		
	Average Value (mg/kg)	SD	*p*
No-AKI	3.9	0.93	0.063
AKI	3.00	0.00	
	Dose of methylprednisolone		
No-AKI	1.28	0.42	0.902
AKI	1.25	0.50	

AKI: Acute Kidney Injury; ASA: Acetylsalicylic Acid.

## Data Availability

The data presented can be provided on request.

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
