# Peer review of "Age-Specific Clinical and Laboratory Features and Renal Involvement in Children with MIS-C: A Single Tertiary Centre Experience from Vojvodina"

_medicina, 2025, doi:10.3390/medicina61071142_

Round 1

Reviewer 1 Report

Comments and Suggestions for Authors

The article is well written, interesting and informative.

There are a few minor revisions suggested:

Line 89: The word ‘findings’ can be removed

Line 128: The p-value quoted in the text for neurological symptoms is different from that in Table 1

Line 137: It sounds more appropriate to say ‘More than half of the participants’ instead of ‘Almost every second participant’ because the authors are referring to 54.6% of the participants

Line 156: pancreatitis or pericarditis? There is no evidence of pancreatitis in the study participants, either from the blood results or abdominal ultrasound, whereas pericarditis was diagnosed in 11 participants

Line 246: pancreatitis or pericarditis?

Line 285: The fatality rate from this study is 1.6%, not 0.64%

Line 305: It would be more appropriate to put ASA in brackets, not acetylsalicylic acid. This can be done in Line 187 when acetylsalicylic acid is first used in the text and subsequently ASA can be used

Reviewer 2 Report

Comments and Suggestions for Authors

Dr. Milanovic et al. analyzed the clinical characteristics of multisystem inflammatory syndrome in children (MIS-C) by age group and identified characteristics of each group. Notably, they also discussed the presumed pathological mechanisms of each characteristic based on previous literature. However, some improvements are needed to enhance the reader's understanding.

Major points

  1. Examination of aspirin dosage and incidence of AKI

The results showed that the aspirin dose was slightly higher in patients with AKI than in non-AKI patients (Table 5). Because the number of patients was small and there was a small difference in the dose of aspirin between group, this result should be interpreted with caution. In addition, it is assumed that the use of aspirin at the onset of AKI and systemic circulatory failure, such as hypotension due to cardiac dysfunction and/or capillary leak from a hyperinflammatory state, occurred simultaneously (Grewal MK et al. Front Pediatr. 2021 Aug 9:9:692256). In light of this, we believe that the discussion of which significantly affected the occurrence of AKI should be reconsidered from a broader perspective.

  1. Risk of renal involvement

   This case series reported only four cases of MIS-C that developed AKI. Additionally, although this paper focuses on renal complications, data on the clinical characteristics of the cases is lacking. If these cases have any commonalities, the main point of this paper will be to highlight them.

  1. Lack of limitation

This paper does not have any limitations. Specifically, it would be better to add the following limitations: this is a retrospective study with a limited number of patients at a single institution, the clinical manifestations and severity of MIS-C vary greatly by race, this study only included patients of similar races, and COVID-19 vaccination status was not taken into account.

Minor point

1, The following terms have not been defined: acute kidney injury (AKI), leukocytosis, lymphopenia, anemia, thrombocytosis, hypoalbuminemia, raised GGT, AST, and ALT levels, and obesity. We recommend indicating reference values in the "Materials and Methods" section.

2, The sentence on line 121 says "n=10/64; 76.9%", but the percentage is incorrect. Please correct it.

3, We understand that normal GGT values vary by age. If there are any additional points to consider, please add them to the discussion.

4, Table 1 shows that there are 11 cases (17.2%) of headaches in the 0-2 age group. The reviewer assumes that describing headaches in this age group is difficult. Please comment on or correct this point.

5, Spelling mistake in the title of Table 1

"MIC-C" ➡ "MIS-C"

Reviewer 3 Report

Comments and Suggestions for Authors

Major Issues to Address

  1. Novelty and Framing of the Study

  2. The novelty is somewhat underplayed. The abstract and introduction should better emphasize what makes this study original (e.g., the lower incidence of AKI compared to other cohorts, or age-specific immune patterns).

  3. Consider including a brief comparison with previously published cohorts in the introduction, especially regarding renal involvement.

  4. Statistical Rigor

  5. While descriptive statistics are comprehensive, there is a need to adjust for confounding variables (e.g., BMI, pre-existing comorbidities) in multivariate models, especially when assessing risk for AKI or organ dysfunction.

  6. Add logistic regression or other multivariate analysis to identify independent predictors of AKI or ICU admission.

  7. Renal Function Discussion

  8. The discussion around proteinuria and AKI is detailed but lacks integration with existing literature on the mechanisms of renal protection in MIS-C.

  9. Address the possible reasons for the low AKI incidence in your cohort compared to others (e.g., earlier treatment initiation, protocol differences).

  10. Clarity on Inclusion Criteria and Diagnostic Accuracy

  11. Explicitly state whether all included patients had positive SARS-CoV-2 tests or only clinical suspicion.

  12. Define the criteria used to classify AKI stages (KDIGO). Was staging performed?

  13. Table Optimization and Supplementary Data

  14. Some tables (particularly Tables 1 and 2) are dense and difficult to follow. Consider simplifying or splitting large tables and moving some to the Supplementary Materials.

  15. Include more graphical representations (e.g., boxplots by age group, heatmaps for organ dysfunction).

Comments on the Quality of English Language
  1. Language and Grammar

  2. While the English is generally acceptable, several grammatical errors and stylistic inconsistencies are present (e.g., “anaemia was more frequent” vs. “anaemia more frequently observed”). A thorough language revision is advised.

  3. Ensure consistent tense usage, especially in the Results and Discussion.

Round 2

Reviewer 2 Report

Comments and Suggestions for Authors

Thank you for your revision work. I am very grateful that you have compiled excellent research results on MIS-C. I think you have clearly answered the reviewers' requests, and the paper has become very good. I have no more comments.

Author Response

Dear Reviewer 2,

Thank You for your comments and all of Your suggestions. It was a pleasure working with You.

Kind regards

Ognjen Ležakov, MD

Reviewer 3 Report

Comments and Suggestions for Authors

Improvements Made in the Second Revision:

  • Statistical clarity: All data across tables and text have been checked and updated for consistency, significance, and appropriateness of tests.

  • Expanded discussion: The revised version thoroughly addresses age-related differences in MIS-C presentations and renal outcomes, including AKI predictors and outcomes in context with recent literature.

  • Clearer conclusions: The manuscript now clearly states that AKI incidence is notably lower than in prior literature, suggesting possible protective factors.

  • Methodological transparency: Inclusion of STROBE adherence, ethics approval, and detailed limitations increase transparency and research quality.

  • Language and structure: English is now more fluent and scientific in tone, with better grammar and structure throughout the article.Final Recommendation:

Accept with minor revisions (language polishing and formatting)

The article is scientifically sound, presents new regional insights into MIS-C (particularly the low AKI incidence), and offers value to the field. Only minor editorial polishing may be required before final publication.

Comments on the Quality of English Language

the English is correct 

Author Response

Dear Reviewer 3,

Thank You for all the suggestions for corrections and all the comments. It was a pleasure working with You. 

Kind regards 

Ognjen Ležakov, MD